# Control of snakebite envenoming: A mathematical modeling study

**Shuaibu Ahijo Abdullahi**[1,3], **Abdulrazaq Garba Habib**[2], **Nafiu Hussaini**[3]*

**1** Department of Mathematics, Modibbo Adama University of Technology, Yola, Adamawa State, Nigeria, **2** Infectious and Tropical Diseases Unit, Department of Medicine, Bayero Univesrity Kano, Aminu Kano Teaching Hospital, Kano, Nigeria, **3** Department of Mathematical Sciences, Bayero University, Kano, Kano State, Nigeria

* nhussaini.mth@buk.edu.ng

## Abstract

A mathematical model is designed to assess the impact of some interventional strategies for curtailing the burden of snakebite envenoming in a community. The model is fitted with real data set. Numerical simulations have shown that public health awareness of the susceptible individuals on snakebite preventive measures could reduce the number of envenoming and prevent deaths and disabilities in the population. The simulations further revealed that if at least fifty percent of snakebite envenoming patients receive early treatment with antivenom a substantial number of deaths will be averted. Furthermore, it is shown using optimal control that combining public health awareness and antivenom treatment averts the highest number of snakebite induced deaths and disability adjusted life years in the study area. To choose the best strategy amidst limited resources in the study area, cost effectiveness analysis in terms of incremental cost effectiveness ratio is performed. It has been established that the control efforts of combining public health awareness of the susceptible individuals and antivenom treatment for victims of snakebite envenoming is the most cost effective strategy. Approximately the sum of US$72,548 is needed to avert 117 deaths or 2,739 disability adjusted life years that are recorded within 21 months in the study area. Thus, the combination of these two control strategies is recommended.

## Author summary

Snakebite envenoming (SBE) is currently one of the life-threatening neglected diseases especially in developing countries. The fight against this menace requires multidisciplinary approach. Owing to significant number of deaths and disabilities recorded per year in West African savanna region, we developed a new mathematical model for SBE in order to gain more insights into the dynamics and control of SBE. It is clear that communities in northeast Nigeria do not have adequate health information on self-protection against SBE and the antivenom is almost scarce and unaffordable. Thus, we evaluated the cost-effectiveness and potential impact of both public health awareness campaign and treatment for SBE as interventional strategies against snakebite. We discovered that public health awareness is crucial in averting SBE, deaths and disabilities. Also, if at least 50% of

**Data Availability Statement:** All relevant data are within the manuscript and its Supporting information files.

**Funding:** SAA would like to acknowledge, with thanks, the support of global Snakebite Initiative

(GSI)/Hamish Ogston Foundation (HOS). While AGH wishes to appreciate the support of African Snakebite Research project Group (ASRG) and Scientific Research Partnership for Next Generation Snakebite Therapies (SRPNTS) supported by National Institutes of Health Research (UK) and Department for International Development (DFID), respectively. The funders had no role in study design, data collection and analysis, decision to publish, or preparation of the manuscript.

**Competing interests:** The authors have declared that no competing interests exist.

SBE victims received treatment within 24 hours of bite, a significant number of deaths and disabilities will be prevented. Furthermore, the study revealed that the combination of public health awareness and treatment decreases the burden of the disease in terms of deaths and disability adjusted life years at a lesser cost as compared with implementing one of these interventions separately. These results can be used as a guide for planning SBE control policy in northeast Nigeria and beyond.

## Introduction

World Health Organization (WHO) defines snakebite envenoming (SBE) as a potentially life threatening disease that typically results from the injection of a mixture of different toxins (venom) following the bite of a venomous snake. SBE typically affects predominantly poor, rural communities in tropical and subtropical countries throughout the world and are significant threats to health and well-being of about 5.8 billion people around the world [1]. Harrison et al., in [2] reported that SBE mainly occurs in penurious settings, where most people engaged in agricultural or pastoral activities, such kind of occupations increase the risk of being bitten by snakes. After many years of neglect, in 2017 WHO reinstated SBE as a priority neglected tropical disease [3, 4]. In an effort to combat the threat caused by SBE in the affected regions worldwide, WHO sets out a plan to reduce snakebite deaths and disabilities by 50% before the year 2030 [1, 5].

One of the major challenges facing the control efforts on reducing SBE in some affected countries is inadequate reliable data. In some countries, the degree of underreporting is more than 70% particularly in rural areas where many snake bite victims use traditional medicine for treatment [6]. Despite the lack of data worldwide, about 4.5—5.4 million people are bitten by snakes every year, out of which 1.8—2.7 million develop envenoming with about 81,000—138,000 deaths. Furthermore, there are 400,000 cases of permanent disability due to this menace [6–9]. In sub-Saharan Africa, where the case of underreporting of data is high, over 250,000 people are reported as being bitten by snakes annually, with an estimated 7,000—20,000 deaths. The under-reporting claim is justified since in West Africa alone, there are 3,557—5,450 deaths that occur yearly. Also at one hospital in Nigeria, 6,687 snakebite cases were treated in just three years, [6, 10]. It has been reported that in sub-Saharan Africa, not less than 6,000 amputations occurred due to snakebite envenoming annually [11]. In Nigeria, Benue valley is the most affected region and has many underreported cases. Furthermore, it has an incidence of 497 per 100,000 population per year with 10—20% mortality [12, 13].

According to Warrell et al. in [14] three snake species carpet viper (Echis ocellatus), black-necked spitting cobra (Naja nigricollis), and puff adder (Bitis arietans) belonging to the Viperidae and Elapidae families are the most significant snakes related with envenoming in Nigeria. Carpet viper account for the majority of the envenoming in Nigeria.

According to WHO in [15], the best effective method of averting snakebite is through educating high-risk communities. Chappuis et al., in [16] recommended an enlightenment campaign to promote the use of protective measures against SBE in snake infested areas. Snake antivenom is considered to be the only treatment that can effectively cure or reverse the effect of snakebite envenoming, however, it may cause adverse reactions as reported in some studies [9, 17–25].

A number of mathematical models have been developed to study the transmission dynamics and control of many neglected zoonotic diseases such as leishmaniasis, rabies, dengue, chagas disease, chikungunya (see for instance, [26–34] and reference therein). Murray in [35],

elucidated that snakebite envenoming shares some epidemiological features with zoonotic diseases. Accordingly, a mathematical model can serve as a tool to study the epidemiology of snakebite in order to gain more insights into its dynamics and control. Unlike other neglected tropical diseases such as dengue, rabies and malaria, to the best of our knowledge, only few research works have been done on mathematical modeling of SBE. Bravo et al, [36] proposed a model using law of mass action to estimate the incidence of snakebite. Also, Kim [37] developed a mathematical model based on the socio-demographic factors that influence mortality risk from SBE in India.

This study extends the above-mentioned models by designing a new mathematical model which incorporates public health awareness campaign as an intervention strategy. The model also includes early and late treatments as well as recovery with or without disabilities. Further, early adverse reaction because of antivenom therapy is considered [20, 24, 25]. It is noteworthy that this study will further assess different control strategies aimed at determining the most cost effective strategy for the control of SBE in the community.

## Materials and methods

### Model formulation

The human population at time t, denoted by $N_H(t)$, is divided into nine mutual exclusive compartments viz: unaware susceptible individuals, $(S_U(t))$, aware susceptible individuals, $(S_E(t))$, SBE individuals, $(I(t))$, individuals receiving early treatment with antivenom, $(T_E(t))$, individuals receiving late treatment with antivenom, $(T_L(t))$, individuals suffering from early adverse reaction (EAR) during early treatment, $(V_E(t))$, individuals suffering from EAR during late treatment, $(V_L(t))$, individuals who recovered with disabilities, $(R_D(t))$, and individuals who recovered without disabilities, $(R_W(t))$. Thus, the total human population is given by

$$N(t) = S_U(t) + S_E(t) + I(t) + T_E(t) + T_L(t) + V_E(t) + V_L(t) + R_D(t) + R_W(t). \qquad (1)$$

The total snake population is represented by $(N_S(t))$. In this work, the aware susceptible individuals referred to those who have received appropriate public health awareness on how to protect themselves against snakebite. On the other hand, the unaware susceptible individuals are those who have not receive the public health awareness and therefore, are not using the protective measures against snakebite.

### Model assumptions

The following are some of the major assumptions made in the construction of the model.

1. Snakebite victim who received treatment with antivenom within 24 hours of bite (envenoming) is considered to be early treatment whereas antivenom administered after 24 hours of bite is regarded as late treatment [19, 38, 39].

2. Early treatment is not associated with death and disability.

3. Recovered individuals are allowed to move into aware susceptible class.

4. It is assumed that the parameter $\theta \in [0, 1]$ measures the effectiveness of the public health awareness in reducing snakebite envenoming in the community. If $\theta = 0$, the public health awareness campaign has no effect on the behavior of susceptible individuals, while if $\theta = 1$ then the public health awareness campaign is 100% effective in improving the behavior of susceptible individuals towards taking protective measures against snakebite.

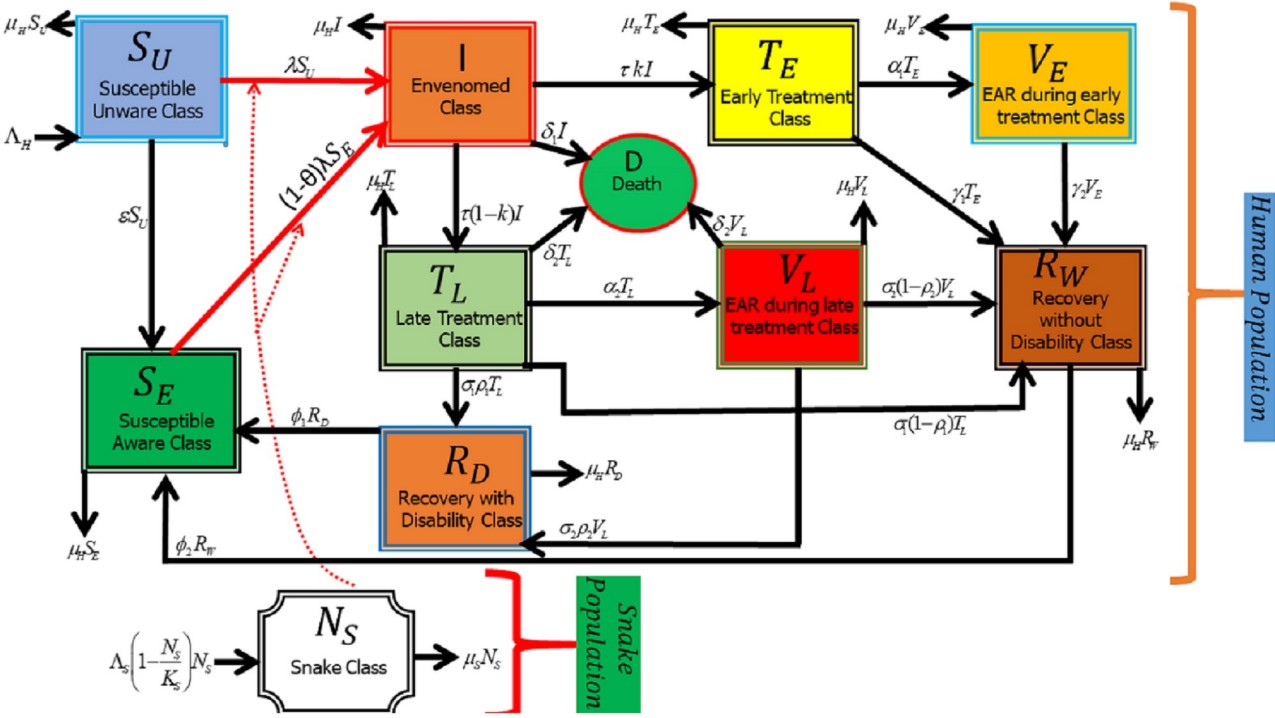

**Fig 1. Schematic diagram of the model.** The diagram describe the movement of individuals from one state to another.

## Model with constant controls

The schematic flow diagram depicted in Fig 1, illustrates the change of state by individuals in the population over time represented by solid lines. Further, it also demonstrates the interaction between humans and snakes in the population which is denoted by dot lines. Using the state variables and parameters of the model presented in Tables 1 and 2, respectively as well as the schematic flow diagram in Fig 1, a deterministic model describing the dynamics of SBE in a given population is established. The model is represented by the system of ordinary differential equations presented below. The complete description of the model is provided in the

**Table 1. Description of state variables of the model.**

| Variables | Description |
|---|---|
| $S_U(t)$ | Unaware susceptible individuals. |
| $S_E(t)$ | Aware susceptible individuals. |
| $I(t)$ | SBE individuals. |
| $T_E(t)$ | Individuals receiving early treatment with antivenom. |
| $T_L(t)$ | Individuals receiving late treatment with antivenom. |
| $V_E(t)$ | Individuals suffering from early adverse reaction during early treatment. |
| $V_L(t)$ | Individuals suffering from early adverse reaction during late treatment. |
| $R_D(t)$ | Individuals who recovered with disabilities. |
| $R_W(t)$ | Individuals who recovered without disabilities. |
| $D(t)$ | cumulative number of deaths due to snakebite. |
| $N_H(t)$ | Total Population of humans. |
| $N_S(t)$ | Population of snakes. |

**Table 2. Description of parameter of the model.**

| Parameter | Description | Units |
|---|---|---|
| $\Lambda_H$ | Recruitment rate of unaware susceptible individuals | $Day^{-1}$ |
| $\Lambda_S$ | Population growth rate of Snakes | $Day^{-1}$ |
| $\mu_H(\mu_S)$ | Natural mortality rates of humans (snakes) | $Day^{-1}$ |
| $\beta$ | Effective snakebite envenomation rate | $Day^{-1}$ |
| $\epsilon$ | Rate of public health awareness campaign | $Day^{-1}$ |
| $\theta$ | Efficacy of public health awareness campaign. | Nil |
| $\tau$ | Rate at which SBE individuals receive treatment with antivenom | $Day^{-1}$ |
| $k$ | Proportion of SBE individuals receiving early treatment with antivenom | Nil |
| $\delta_i(i=1,2)$ | SBE induced death rates in $I$, and $T_L$ and $V_L$ compartments respectively | $Day^{-1}$ |
| $\alpha_i(i=1,2)$ | Rate at which individuals receiving treatment with antivenom suffer from EAR in $T_E$ and $T_L$ compartments respectively | $Day^{-1}$ |
| $\gamma_i(i=1,2)$ | Recovery rate without disability of individuals in $T_E$ and $V_E$ compartments | $Day^{-1}$ |
| $\sigma_i(i=1,2)$ | Recovery rate with disability of individuals in $T_L$ and $V_L$ compartments | $Day^{-1}$ |
| $\rho_i(i=1,2)$ | Proportions of individuals who recovered with disabilities in $T_L$ and $V_L$ compartments respectively | Nil |
| $\phi_i(i=1,2)$ | Transition rates of individuals in $R_D$ and $R_W$ compartments to $S_E$ compartment | $Day^{-1}$ |
| $K_S$ | Carrying capacity of snake | Nil |

supplementary material S1 File.

$$\frac{dS_U}{dt} = \Lambda_H - (\lambda + K_1)S_U,$$

$$\frac{dS_E}{dt} = \epsilon S_U + \phi_1 R_D + \phi_2 R_W - (\Pi_1\lambda + K_2)S_E,$$

$$\frac{dI}{dt} = (\Pi_1 S_E + S_U)\lambda - K_3 I,$$

$$\frac{dT_E}{dt} = \tau k I - K_4 T_E,$$

$$\frac{dT_L}{dt} = \tau\Pi_2 I - K_5 T_L,$$

$$\frac{dV_E}{dt} = \alpha_1 T_E - K_6 V_E,$$

$$\frac{dV_L}{dt} = \alpha_2 T_L - K_7 V_L,$$

$$\frac{dR_D}{dt} = \sigma_1\rho_1 T_L + \sigma_2\rho_2 V_L - K_8 R_D,$$

$$\frac{dR_W}{dt} = \gamma_1 T_E + \gamma_2 V_E + \sigma_1\Pi_3 T_L + \sigma_2\Pi_4 V_L - K_9 R_W,$$

$$\frac{dN_S}{dt} = \Lambda_S N_S\left(1 - \frac{N_S}{K_S}\right) - \mu_S N_S,$$

$$\frac{dD}{dt} = \delta_1 I + (T_L + V_L)\delta_2,$$

$$\lambda(t) = \frac{\beta N_S}{N_H + N_S},$$

(2)

where,

$$K_1 = \epsilon + \mu_H, \quad K_2 = \mu_H, \quad K_3 = \tau + \delta_1 + \mu_H, \quad K_4 = \alpha_1 + \gamma_1 + \mu_H,$$
$$K_5 = \alpha_2 + \sigma_1 + \delta_2 + \mu_H, \quad K_6 = \gamma_2 + \mu_H, \quad K_7 = \sigma_2 + \delta_2 + \mu_H, \quad K_8 = \phi_1 + \mu_H,$$
$$K_9 = \phi_2 + \mu_H, \quad \Pi_1 = 1 - \theta, \quad \Pi_2 = 1 - k, \quad \Pi_3 = 1 - \rho_1, \quad \Pi_4 = 1 - \rho_4,$$

subject to the initial conditions

$$S_U(0) > 0, \quad S_E(0) \geq 0, \quad I(0) \geq 0, \quad T_E(0) \geq 0, \quad T_L(0) \geq 0, \quad V_E(0) \geq 0,$$
$$V_L(0) \geq 0, \quad R_D(0) \geq 0, \quad R_W(0) \geq 0, \quad N_S(0) \geq 0, \quad D(0) \geq 0. \tag{3}$$

## Model with time dependent controls

An optimal control problem is developed by incorporating the following time dependent control strategies into the constant control model given in Eq (2):

1. $u_1(t)$ with $0 \leq u_1(t) \leq 1$ represents the control effort on educating the susceptible individuals on the risk associated with SBE. This control strategy promotes the use of protective measures such as hand gloves, boots, long sleeves wear etc.

2. $u_2(t)$ with $0 \leq u_2(t) \leq 1$ is the control effort aimed at treating the SBE individuals with antivenom.

Two time dependent control variables are introduced to seek for the optimal result with least effort required to curtail the burden of SBE in the population at a minimum cost of implementation. Therefore, the optimal control model is given by

$$\frac{dS_U}{dt} = \Lambda_H - (\lambda + \epsilon u_1(t) + \mu_H)S_U,$$
$$\frac{dS_E}{dt} = \epsilon u_1(t)S_U + \phi_1 R_D + \phi_2 R_W - ((1-\theta)\lambda + \mu_H)S_E,$$
$$\frac{dI}{dt} = ((1-\theta)S_E + S_U)\lambda - (\tau u_2(t) + \delta_1 + \mu_H)I,$$
$$\frac{dT_E}{dt} = \tau u_2(t)kI - (\alpha_1 + \gamma_1 + \mu_H)T_E,$$
$$\frac{dT_L}{dt} = \tau u_2(t)(1-k)I - (\alpha_2 + \sigma_1 + (1 - u_2(t))\delta_2 + \mu_H)T_L,$$
$$\frac{dV_E}{dt} = \alpha_1 T_E - (\gamma_2 + \mu_H)V_E,$$
$$\frac{dV_L}{dt} = \alpha_2 T_L - (\sigma_2 + (1 - u_2(t))\delta_2 + \mu_H)V_L, \tag{4}$$
$$\frac{dR_D}{dt} = \sigma_1 \rho_1 T_L + \sigma_2 \rho_2 V_L - (\phi_1 + \mu_H)R_D,$$
$$\frac{dR_W}{dt} = \gamma_1 T_E + \gamma_2 V_E + \sigma_1 \Pi_3 T_L + \sigma_2 \Pi_4 V_L - (\phi_2 + \mu_H)R_W,$$
$$\frac{dN_S}{dt} = \Lambda_S N_S \left(1 - \frac{N_S}{K_S}\right) - \mu_S N_S,$$
$$\frac{dD}{dt} = \delta_1 I + (T_L + V_L)(1 - u_2(t))\delta_2,$$
$$\lambda(t) = \frac{\beta N_S}{N_H + N_S},$$

with the initial conditions given by Eq (3).

To explore the optimal level of efforts that would be required to control snakebite envenoming in the study area, we constructed an objective functional $J(u_1, u_2)$, whose goal is to minimize the number of snakebite envenoming individuals at time $t$, given by $I(t)$, the cumulative number of snakebite induced death at time $t$, denoted by $D(t)$ and the costs of applying the control efforts, $u_1$ and $u_2$, on public health education campaign for susceptible individuals and treatment of envenomed victims, respectively. In line with Rodrigues et al., [40], Agusto et al., [41], we used a quadratic cost functional with respect to the control variables $u_1$ and $u_2$ in order to guarantee convexity condition for optimality mentioned in Colaneri et al., [42]. Thus, the objective functional corresponding to the optimal control model in Eq (4) is given by

$$J(u_1, u_2) = \int_0^T \left( B_1 I(t) + B_2 D(t) + \frac{1}{2} \sum_{i=1}^{2} \left( C_i u_i^2 \right) \right) dt, \tag{5}$$

subject to the state system given by Eq (4). The goal is to minimize the number of SBE and death induced by same in the population at a minimal cost of implementing the control measures. In Eq (5), the quantities $B_1$ and $B_2$ are the weight constants corresponding to the population of SBE individuals and cumulative death induced by the disease respectively. While the quantities, $C_1$ and $C_2$ are the relative costs weight constants for the controls $u_1$, and $u_2$ respectively. We assume that the cost of each control is proportional to the square of its associated control function. The term $\frac{C_1 u_1^2}{2}$ is the cost corresponding to the control effort on public health education of susceptibles on the risk associated with snakebite and the promotion of the use of protective measures. Similarly, $\frac{C_2 u_2^2}{2}$ is the cost associated with the control effort on treating SBE patients. Note that the square of the controls indicates the non-linearity of cost function while the half-term minimizes the effect of applying the controls.

Our aim is to search for the controls functions $(u_1^*, u_2^*)$ such that

$$J(u_1^*, u_2^*) = \min\{(u_1, u_2) | (u_1, u_2) \in \Psi\}, \tag{6}$$

where

$\Psi = \{(u_1, u_2) | u_i(t) \text{ is Lebesgue measurable on } [0, T], 0 \leq u_i(t) \leq 1, i = 1, 2\}$ is the control set of system Eq (4). The existence of the optimal controls and the derivation of the optimality system is reported in the supplementary S2 File.

## Study area

This research focused on the northeast Nigeria which comprises six states that include Adamawa, Bauchi, Borno, Gombe, Taraba and Yobe as shown in Fig 2. According to National Bureau of Statistics (NBS) in [43] the projected total population of the six states in the northeast Nigeria is 26,263,866. Majority of people in this region engaged in agricultural activities such as farming, livestock rearing, and fishing. These activities placed them at high risk of snakebite. This region harbors some highly medically important snakes like carpet viper, black-necked spitting cobra, and puff adder. Snakebite Treatment and Research Hospital (STRH) is located in Kaltungo local government area in Gombe state which makes snakebite treatment accessible to people in the region. Kaltungo is one of the snakebite hot spots in Nigeria.

## Snakebite data collection

Data on snakebite is primarily collected from STRH in Gombe state, for the period of twenty one months from January, 2019 to September, 2020. These data include the number of SBE individuals, the number of individuals receiving early treatment with antivenom, the number

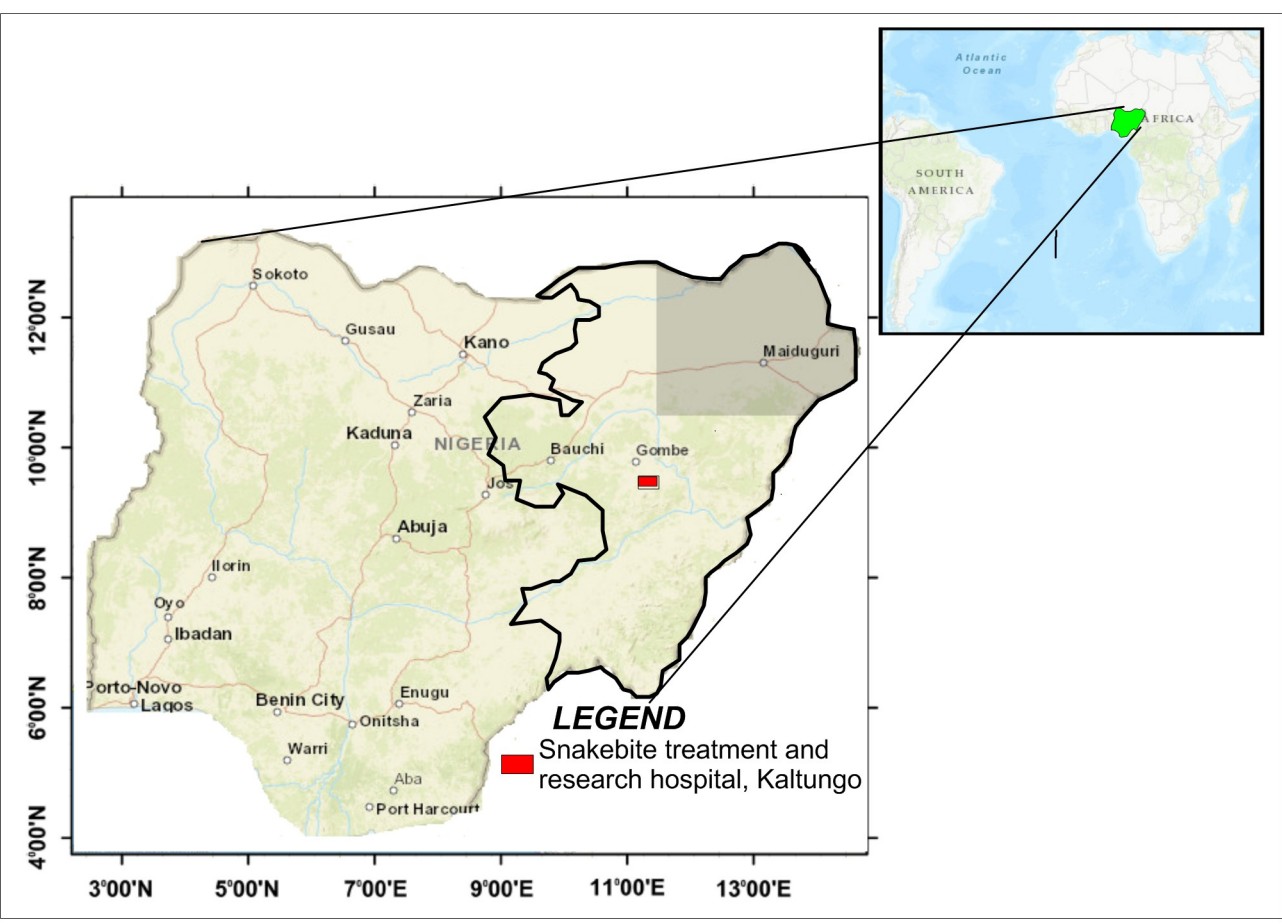

**Fig 2. Map of Nigeria showing the study area.** The map portrays the states in the study area and the regional snakebite treatment center, Kaltungo. The map was extracted from public domain map: https://pubs.er.usgs.gov/publication/ofr7261 and modified using Golden Software Surfer 11.0.642.

individuals receiving late treatment with antivenom, the number of individuals suffering from EAR during late treatment,the number of individuals suffering from EAR during early treatment, the number of individuals recovered with disability, number of individuals recovered without disability, and the number of snakebite deaths as presented in supplementary S1 Table. The cumulative number of these seven different sets of data on snakebite cases will be used to fit the model as well as to performed some numerical simulations. Thus, the data used in this work are aggregated from all the six states that made up the study area. Note that all the seven sets of snakebite data collected are due to saw scaled viper (carpet viper).

## Model fitting and parameters estimation

Real data collected from STRH is used to estimate the unknown parameters as well as to fit the model with the monthly reported data. The data is presented in supplementary S1 Table. The human and snake demographic parameters $\mu_H$, $\Lambda_H$ and $\mu_S$ are parameterized as follows:

- The total population of the study area is 26,263,866 [43].

- Life expectancy of Nigeria as at 2018 is 54.332 years [44], $\mu_H = \frac{1}{54.332}$, thus, $\mu_H = 5.04 \times 10^{-6}$ per day.

- Using the relation, $\frac{\Lambda_H}{\mu_H} = 26,263,866$, it follows that $\Lambda_H = 1324$ per day.

- The life expectancy of Saw-scaled viper is 12 years [45], so that $\mu_S = \frac{1}{12}$ and hence $\mu_S = 2.283 \times 10^{-4}$ per day.

Following Zu et al. [46], all other parameter values and the initial conditions of state variables in the model are estimated using the least square method and Markov Chain Monte Carlo (MCMC) technique. A set of results is estimated using the least square method with 100,000 number of iterations and the outcome is employed as initial guess for the MCMC method. To ensure the convergence of MCMC algorithm we used Gelman-Rubin diagnostic test implemented in MATLAB. We set the number of iteration to be 80000 with a burn-in of 40000 iterations. According to Gelman and Rubin [47], if chains have converged to the target posterior distribution, then Potential Scale Reduction Factor (PSRF) denoted by $R_c$ should be sufficiently close to 1. The result in Table 3 shows that the $R_c$ values are between 0.99 to 1.04 and thus all the chains have converged.

The estimated initial conditions and values of the parameters in the model are presented in Tables 4 and 5, respectively. The comparison between the estimated values by model and the real reported monthly data are depicted in Fig 3. The estimated outcomes of the model are in good agreement with the actual reported data. Therefore, the proposed model and the estimated parameter values can be used to predict the SBE incidence as well as understanding its dynamics in Nigeria and beyond.

## Estimation of cost of public health enlightenment campaign

We estimated per capita cost of public health awareness on the risk associated with SBE and its preventive measures in the study area. Broadcasting media and mobile technology are considered. From Tables 6 and 7, the total cost of enlightenment is US$7,332,887.23 and the total

**Table 3. The report of Gelman-Rubin diagnostics Test for MCMC.**

| Name of Parameter | Values of Potential Scale Reduction Factor ($R_C$) |
|---|---|
| $\beta$ | 1.0017 |
| $\epsilon$ | 0.9973 |
| $\theta$ | 1.0171 |
| $\tau$ | 1.0363 |
| $\delta_1$ | 0.9971 |
| $\delta_2$ | 0.9986 |
| $\gamma$ | 1.0113 |
| $\sigma$ | 0.9979 |
| $\alpha_1$ | 1.0086 |
| $\alpha_2$ | 1.0089 |
| $\rho_1$ | 0.9985 |
| $\rho_2$ | 1.0013 |
| $A_S$ | 1.0001 |
| $\phi_1$ | 0.9950 |
| $\phi_2$ | 1.0004 |
| $k$ | 0.9991 |
| $S_U(0)$ | 1.0056 |
| $S_E(0)$ | 1.0260 |
| $N_S(0)$ | 1.0027 |
| $K_S$ | 1.0048 |

**Table 4. Estimated initial condition of state variables.**

| Variables | Baseline | 95% Confidence Interval | Reference |
|---|---|---|---|
| $S_U(0)$ | $2.1459 \times 10^7$ | $2.1097 \times 10^7$, $2.1820 \times 10^7$ | Estimated |
| $S_E(0)$ | $6.5132 \times 10^3$ | $(6.3876 \times 10^3, 6.6388 \times 10^3)$ | Estimated |
| $I(t)$ | 99 | - | Reported data |
| $T_E(0)$ | 76 | - | Reported data |
| $T_L(0)$ | 7 | - | Reported data |
| $V_E(0)$ | 8 | - | Reported data |
| $V_L(0)$ | 0 | - | Reported data |
| $R_D(0)$ | 4 | - | Reported data |
| $R_W(0)$ | 93 | - | Reported data |
| $N_S(0)$ | $1.2250 \times 10^4$ | $(1.2179 \times 10^4, 1.2320 \times 10^4)$ | Estimated |
| $D_S(0)$ | 2 | - | Reported data |

population of the study area is 26,263,866 [43]. Thus, per capita cost of public heath enlightenment is US$0.28.

## Results and discussion

### Numerical assessment of impact of public health awareness

Let us consider the following three different scenarios of applying the public health awareness using the model with constant controls:

**Table 5. Estimated values of the model parameter.**

| Parameter | Baseline | 95% Confidence interval | Units | Reference |
|---|---|---|---|---|
| $\Lambda_H$ | 1324 | - | $Day^{-1}$ | [43, 44] |
| $\Lambda_S$ | 0.1925 | $(0.1924, 0.1926)$ | $Day^{-1}$ | Estimated |
| $\mu_H$ | $5.04 \times 10^{-6}$ | - | $Day^{-1}$ | [44] |
| $\mu_S$ | $2.283 \times 10^{-4}$ | - | $Day^{-1}$ | [45] |
| $\beta$ | 0.0742 | $(0.0741, 0.0743)$ | $Day^{-1}$ | Estimated |
| $\epsilon$ | 0.0051 | $(0.0049, 0.0053)$ | $Day^{-1}$ | Estimated |
| $\theta$ | $1.7729 \times 10^{-4}$ | $(0.524 \times 10^{-4}, 3.022 \times 10^{-4})$ | Nil | Estimated |
| $\tau$ | 0.9997 | $(0.9994, 1.0000)$ | $Day^{-1}$ | Estimated |
| $k$ | 0.8073 | $(0.8070, 0.8076)$ | Nil | Estimated |
| $\delta_1$ | 0.0025 | $(0.0023, 0.0028)$ | $Day^{-1}$ | Estimated |
| $\delta_2$ | $4.2564 \times 10^{-4}$ | $(2.462 \times 10^{-4}, 6.05 \times 10^{-4})$ | $Day^{-1}$ | Estimated |
| $\alpha_1$ | 0.1215 | $(0.1214, 0.1216)$ | $Day^{-1}$ | Estimated |
| $\alpha_2$ | 0.1708 | $(0.1706, 0.1709)$ | $Day^{-1}$ | Estimated |
| $\gamma_i(i = 1, 2)$ | 0.9310 | $(0.9307, 0.9313)$ | $Day^{-1}$ | Estimated |
| $\sigma_i(i = 1, 2)$ | 0.9924 | $(0.9898, 0.9950)$ | $Day^{-1}$ | Estimated |
| $\rho_1$ | 0.1500 | $(0.1497, 0.1503)$ | Nil | Estimated |
| $\rho_2$ | 0.9985 | $(0.9982, 0.9987)$ | Nil | Estimated |
| $\phi_1$ | 0.5233 | $(0.5231, 0.5234)$ | $Day^{-1}$ | Estimated |
| $\phi_2$ | 0.9416 | $(0.9384, 0.9448)$ | $Day^{-1}$ | Estimated |
| $K_S$ | $6.6604 \times 10^4$ | $(6.6457 \times 10^4, 6.6752 \times 10^4)$ | Nil | Estimated |
| $B_i(i = 1, 2)$ | 1 | - | Nil | Assumed |
| $C_1$ | 0.28 | - | US$ | Estimated |
| $C_2$ | 237 | - | US$ | [48–50] |

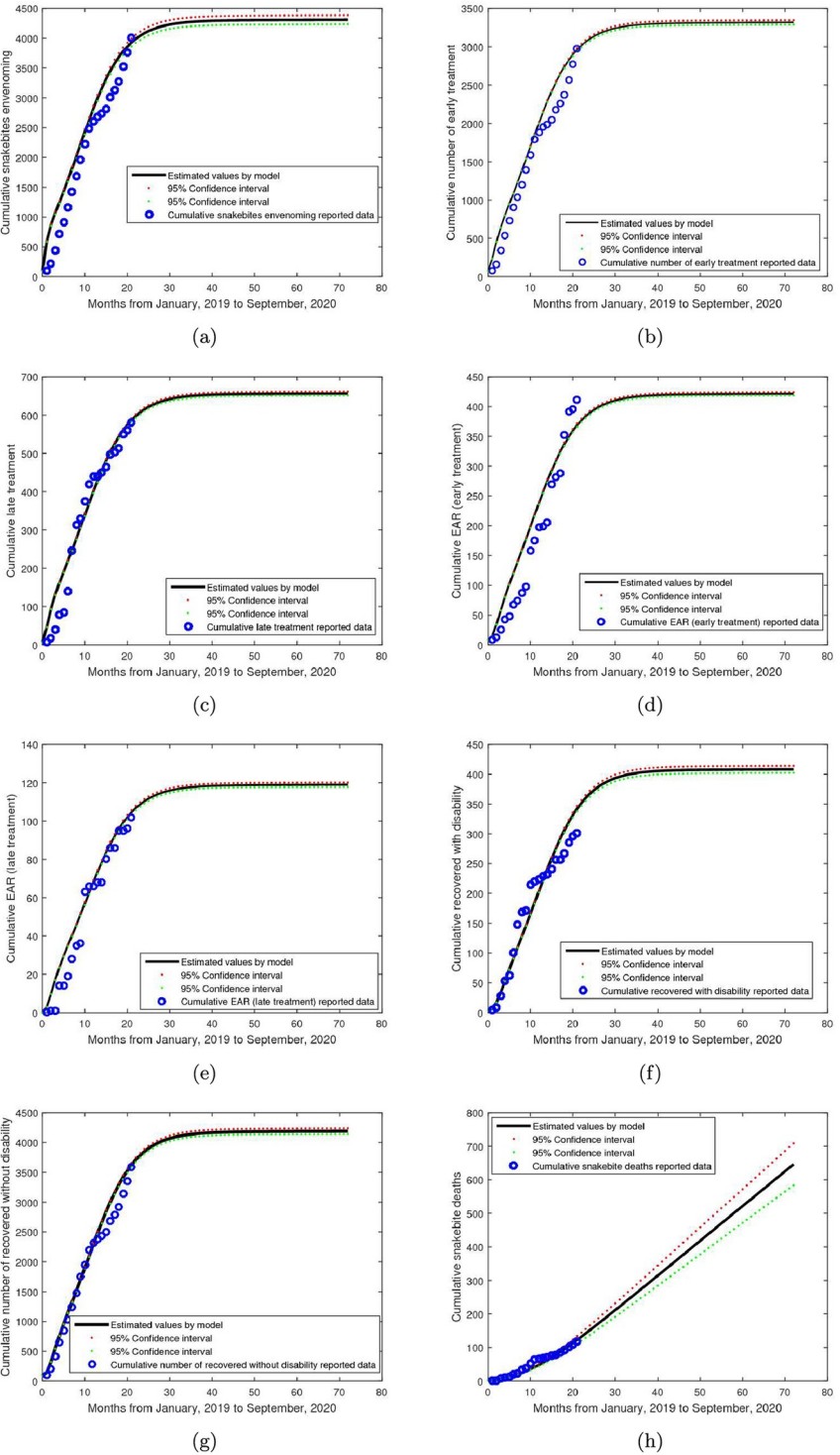

**Fig 3. Graphs showing the results of the model fitting with the reported data.** The model with constant controls is fitted with the reported cumulative number of (a) snakebite envenoming (b) early treatment (c) late treatment (d) EAR during early treatment (e) EAR during late treatment (f) recovered with disability (g) recovered without disability and (h) snakebite deaths. It can be seen that the model fitted well with the respective data sets collected from the snakebite treatment and research hospital, Kaltungo.

**Table 6. Estimation of cost of public health awareness on snakebite in 6 states of North East (NE) Nigeria using Broadcasting media.**

| Activity | Resources needed | Amount($) | Reference |
|---|---|---:|---:|
| Jingle Production in 7 Languages. | Audio | 531.96 | Reported |
| Jingle Production in 7 Languages. | Video | 2,127.82 | Reported |
| Airing of jingle in Adamawa | Radio stations | 82,073.12 | Reported |
| | Television stations | 20,974.24 | Reported |
| Airing of jingle in Bauchi | Radio stations | 61,554.84 | Reported |
| | Television stations | 20,974.24 | Reported |
| Airing of jingle in Borno | Radio stations | 61,554.84 | Reported |
| | Television stations | 20,974.24 | Reported |
| Airing of jingle in Gombe | Radio stations | 61,554.84 | Reported |
| | Television stations | 20,974.24 | Reported |
| Airing of jingle in Taraba | Radio stations | 41,036.56 | Reported |
| | Television stations | 41,948.48 | Reported |
| Airing of jingle in Yobe | Radio stations | 41,036.56 | Reported |
| | Television stations | 41,948.48 | Reported |
| Total | | 444,562.66 | |

**Table 7. Estimation of cost of public health awareness on snakebite in six states of NE Nigeria using mobile phone.**

| Activity | unit cost($) | Population | Frequency | No of Months | Amount($) |
|---|---|---|---|---|---|
| Bulk SMS | 0.0051 | 14,069,290.38 | 8 | 12 | 6,888,324.57 |

*CaseI*: Low level of public health awareness campaign coverage and its efficacy (i.e. $\epsilon = \theta = 10\%$).

*CaseII*: Moderate level of public health awareness campaign coverage and its efficacy (i.e. $\epsilon = \theta = 50\%$).

*CaseIII*: High level of public health awareness campaign coverage and its efficacy (i.e. $\epsilon = \theta = 90\%$).

The health benefits used for assessing the impact and effectiveness of the public health awareness campaign are the number of SBE, death and disability averted. The results in Table 8 and Fig 4 show that an increase in public health awareness increases the number of SBE averted cases as depicted in Fig 4A. In addition, the results further show that more number of death and disability are prevented when such intervention is increased in terms of coverage and efficacy (see Fig 4B and 4C). This outcome suggests that public health advocacy could serve as a strong non-pharmaceutical control measure of reducing the number of SBE, death and disability in the region.

**Table 8. Simulations showing the impact and effectiveness of public health awareness over the period of 12 months.**

| Cases | SBE averted | Death averted | Disability averted |
|---|---|---|---|
| *Case I* | 1,120 | 10 | 57 |
| *Case II* | 10,515 | 111 | 609 |
| *Case III* | 20,075 | 223 | 1,200 |

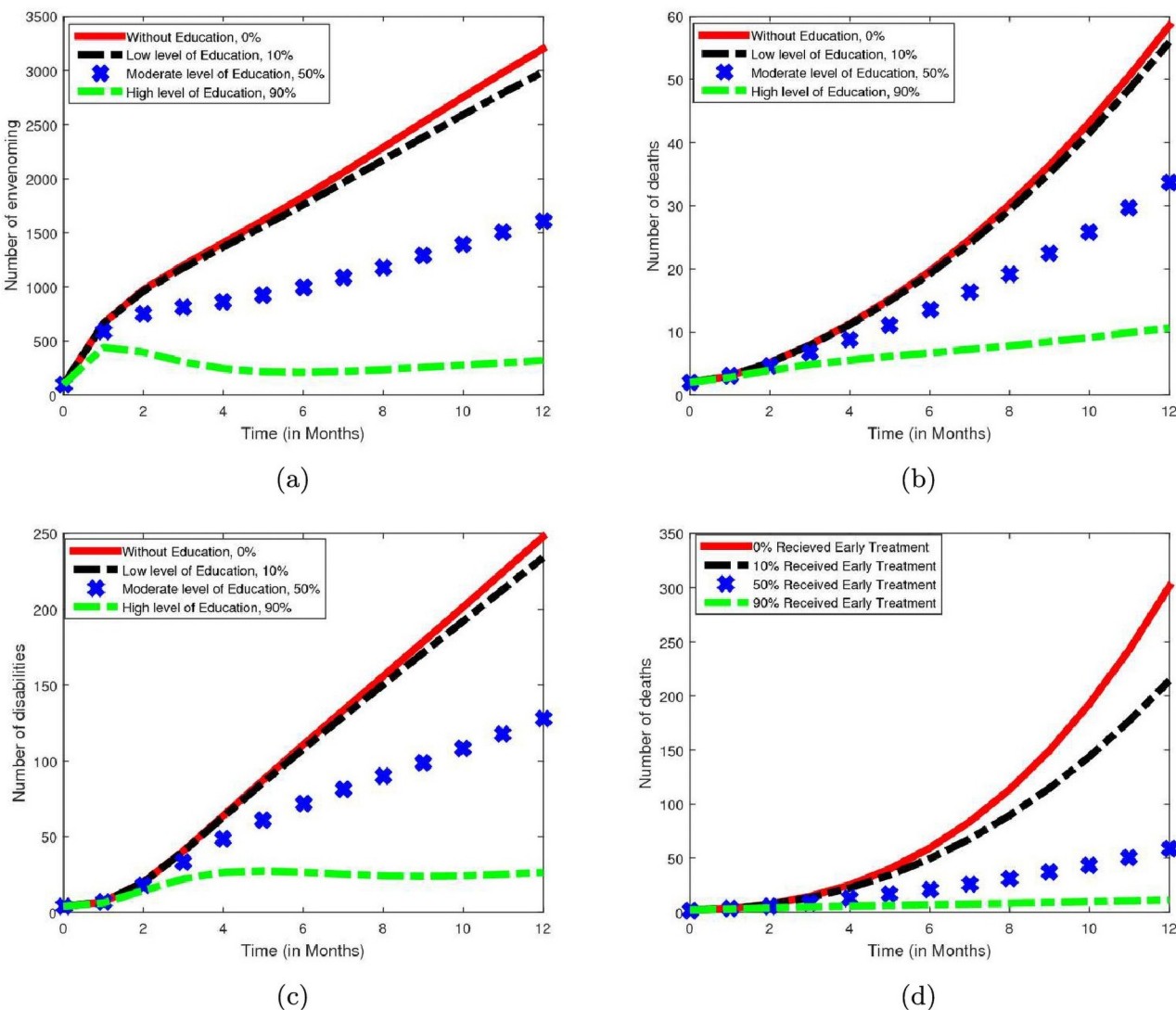

**Fig 4. Simulations showing the impact and effectiveness of public health awareness and early treatment.** The graphs clearly show the impact and effectiveness of public health awareness in averting number of (a) snakebite envenoming (b) snakebite induced death (c) disability. While (d) shows the impact of early treatment with antivenom on snakebite induced death. In Fig 4A, 4B and 4C the red, black, blue and green colors correspond to 0%, 10%, 50% and 90% public health awareness campaign coverage and its efficacy, respectively. In Fig 4D, the red, black, blue and green colors represent 0%, 10%, 50% and 90% of envenoming individuals receiving early treatment with antivenom.

### Numerical assessment of impact of early treatment

Using the model with constant controls, numerical simulation is performed to appraise the potential impact of early treatment on the dynamics of SBE in terms of deaths averted for the period of one year, by considering the following scenario:

*case I*: 10% of SBE individuals receive early treatment (i.e. $k = 10$%).

*case II*: 50% of SBE individuals receive early treatment (i.e. $k = 50$%).

*case III*: 90% of SBE individuals receive early treatment (i.e. $k = 90$%).

The number of death averted corresponding to 10%, 50% and 90% of SBE victims that receive early treatment are 296, 918 and 1,146, respectively. This result indicates that an

increase in proportion of individuals receiving early treatment increases the number of death averted over the period of time under study. In addition, the outcome depicted in Fig 4D illustrates that seeking for early treatment when snakebite occurs is very significant in reducing the number deaths due to SBE in the study area. It is observed that when more than 50% of SBE victims receive early treatment then scores of deaths will be prevented. This outcome suggests that even with adequate supply of effective and affordable antivenom as proposed by WHO in the road map to reduce snakebite mortality by 50% before the year 2030, if not administer at the right time, might not be able to reduce the death by half to meet WHO target. Thus, educating the risk population to seek for early treatment is also essential in achieving the set objectives.

## Procedure for solving optimality system

In order to obtain solution for the optimality system which consists of state equations Eq (4), adjoint system in S2 File, the characterizations in S2 File and corresponding initial/final conditions, we apply the Runge-Kutta fourth order technique which is more accurate. It is a multiple-step method and also known as forward-backward sweep method (for detail description of this technique see [51]). The procedure starts with an initial guess on the control variable given initial conditions for the state variables, the solutions for the state equations will be approximated using the Runge-Kutta forward sweep technique. Given the state solutions from the preceding step and the final time conditions for adjoints, the solutions for adjoint equations will then be approximated using Runge-Kutta backward sweep method. The value of control variables is updated by taking the average of the preceding value and the new value arising from the control characterization. The procedure is repeated for forward numerical scheme and updating the controls until successive values of all states, adjoints, and controls converge.

## Numerical simulation of the optimal control model

The numerical solutions of the consequential optimality system obtained in S2 File are carried out. The forward-backward sweep method is employed using the initials conditions and parameter values in Tables 4 and 5, respectively, with ($\sigma_1 = \sigma_2 = 0.45$, $\epsilon = \theta = 0.65$, $k = 0.55$). The algorithm starts with an initial guess of ($u_1$, $u_2$) = (0, 0) for the optimal controls and the state variables are then solved forward in time using Runge-Kutta method of the fourth order. Further, the state variables and initial control guess are used to solve the adjoint equations in S2 File backward in time with the given final condition in S2 File, using the backward fourth order Runge-Kutta method. The controls $u_1$ and $u_2$ are then updated and used to solve the state equations and then the adjoint system. This iterative process ends when the solutions converge. The simulations are carried-out over the period of 12 months. In order to demonstrate the effect of the implementation of the time dependent controls, the following strategies are considered:

*Strategy A*: public health awareness (i.e., $u_1$) only,

*Strategy B*: treatment of SBE victims (i.e., $u_2$) only,

*Strategy C*: combination of the strategies A and B (i.e., $u_1$ and $u_2$).

Fig 5 shows the impact of implementing the control strategies on the dynamics of SBE for the period of one year in the northeast Nigeria. It is observed that, as shown in Fig 5A, 5B and 5C, at the beginning of the first three months, the impact of the control strategies implemented either independently or simultaneously are insignificant. After this period a significant

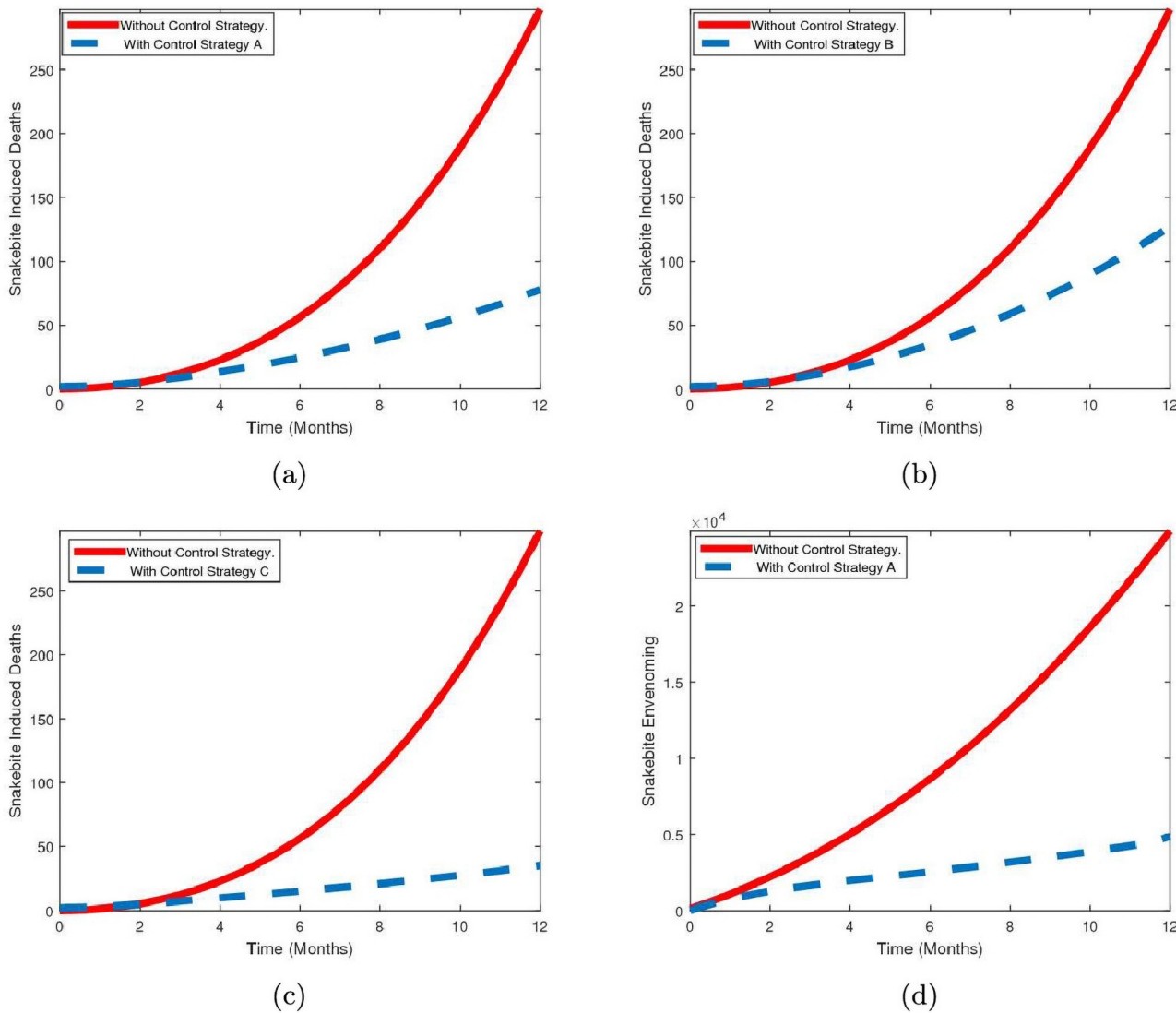

**Fig 5. Simulation results of the optimal control model.** The simulation results of the optimal control model showing the effectiveness of (a) strategy A in averting the number of snakebite induced deaths as against without strategy, (b) strategy B in averting the number of snakebite induced deaths as against without strategy, (c) strategy C in averting the number of snakebite induced deaths as against without strategy, (d) strategy A in averting number of snakebite envenoming as against without strategy. In the graphs the solid line denotes without strategy while dotted line indicates with control strategy.

reduction in the number of snakebite induced deaths is noticed. Further, implementation of strategy C averts more deaths followed by strategy A. It is noteworthy that whenever any of these control strategies is implemented, it has to be maintained over the planning horizon in order to control the number of deaths. On the other hand, in the absence of control measures, the rate of death is significantly high. Fig 5D, presents the effect of implementing control strategy A on snakebite envenoming.

It is shown in S1 Fig, that the optimal solution is attained when the control effort on public health enlightenment is firmly observed at maximum level from the onset and should be maintained for approximately the period of 11.16 months before relaxing the control effort to the barest minimum. In case of the control effort on the treatment of SBE patients, the optimal solution is attained when the control effort is rapidly increased to reach the maximum level at

0.12 months and is maintained for the period of 11.88 months before reducing it to zero. Consequently, to reduce the burden of SBE in terms of deaths avertion within the planning horizon of one year, these control efforts must be maintained at a maximum level.

## Cost-effectiveness analysis

In this section, we compared the costs and benefits of the different control strategies employed to avert snakebite induced death with particular reference to northeast Nigeria. In recent times, cost effectiveness analysis has become an important tool to many researchers especially in the field of mathematical epidemiology see for instance [52–55]. For effective allocation of resources to control snakebite cases, public health decision makers need to know the impact and cost-effectiveness of snakebite prevention and treatment programmes. In order to choose the right intervention policy, a cost-effectiveness ratio (CER) in terms of incremental cost-effectiveness ratio (ICER) is calculated. Furthermore, the effectiveness of an intervention is measured in terms of Quality Adjusted Life Years (QALYs), deaths prevented or infections averted. In this study, snakebite induced death and Disability Adjusted Life Year (DALY) averted are employed as health benefit of the control interventions. In line with Hove-Musekwa et al., [52] and Adamu et al., [55], a linear cost function with respect to the control variables $u_1$ and $u_2$ of the cost effectiveness analysis for the control strategies is used. The total discounted cost function for control strategy i, is given by;

$$Cost_i = \int_0^T (C_1 u_1 \epsilon S_U + C_2 u_2 (kT_E + (1-k)T_L)\tau + \alpha_1 V_E + \alpha_2 V_L)e^{-rt}dt, \qquad (7)$$

where $r = 5\%$ is a discount rate [52] and $i = A, B, C$. Following Weinstein [56], the formula for computing ICER for two competing strategies I and J is given by

$$ICER_{I,J} = \frac{Cost_I - Cost_J}{Death\ Averted_I - Death\ Averted_J} \qquad (8)$$

Using Eq (8) the ICER for strategies A, B and C are computed as follows:

$$ICER_A = \frac{6.4615 \times 10^6}{5.8570 \times 10^3} = 1.1032 \times 10^3,$$

$$ICER_{B,A} = \frac{(3.0030 - 6.4615) \times 10^6}{(4.3161 - 5.8570) \times 10^6} = 2.2445 \times 10^3,$$

$$ICER_{C,B} = \frac{(7.3100 - 3.0030) \times 10^6}{(7.2254 - 4.3161) \times 10^6} = 1.4804 \times 10^3.$$

The results of the ICER for strategies A, B and C are presented in Table 9. Comparing strategy A and strategy B, it is obvious that the $ICER_A$ is less than $ICER_{B,A}$. This shows that strategy B is less effective than strategy A, meaning that strategy B is dominated. Therefore, strategy B is

**Table 9. ICER of control strategies in the order of death averted.**

| Strategy | Death averted | Cost of strategy($) | ICER/($) |
|---|---|---|---|
| Strategy A | 5,857 | 6,461,500 | 1,103.2 |
| Strategy B | 4,316.1 | 3,003,000 | 2,244.5 |
| Strategy C | 7,225.4 | 7,310,000 | 1,480.4 |

**Table 10. Comparison between ICER of strategies A and C.**

| Strategy | Deaths averted | Cost of strategy($) | ICER/($) |
|---|---|---|---|
| Strategy A | 5,857 | 6,461,500 | 1,103.2 |
| Strategy C | 7,225.4 | 7,310,000 | 620.07 |

removed from the list. Accordingly, the ICER of strategies A and C are evaluated using analogous technique and the result is presented in Table 10.

The result in Table 10 shows that strategy A is dominated by strategy C because $ICER_{C,A}$ is less than $ICER_A$. This suggests that strategy A is more costly and less effective than strategy C. Therefore, implementation of control efforts on public health awareness and treatment simultaneously is the most cost-effective strategy. This strategy is capable of averting more number of deaths at a lesser cost of implementation. Using the criteria for choosing cost effectiveness threshold based on per capita Gross Domestic Product (GDP) established in [57], strategy C is highly cost effective in the region, because its ICER is less than threefold per capita GDP of Nigeria. According to [58], the estimated per capita GDP of Nigeria as at 2019 is $2,229.9. Following this criteria, strategy A is just cost effective since its ICER per death averted is less than twofold per capita GDP. Thus, strategy C is recommended because of its capability of averting highest number deaths at a lesser cost. The criteria for selecting cost effectiveness threshold of strategy based on per capita GDP of a region or country for developing countries is presented as follows. A strategy is considered to be:

1. highly cost effective if the ICER is less than one times per capita GDP;

2. cost effective if the ICER is between one times per capita GDP and less than threefold per capita GDP;

3. not cost effective if the ICER is greater than threefold per capita GDP.

Following Habib, et al., [49] and Hamza et al., [50], 23.41 discounted DALYs approximation is equivalent to one early mortality due to snakebite. Therefore, the deaths averted by each strategy shown in Table 9, column 2, were converted to DALYs by taking the product of the total number of deaths averted by each strategy and 23.41 DALYs and the results are shown in Table 11 column 2. Consequently, we computed the ICER in terms of DALY averted as health benefit yielding a cost/DALY averted of $95.88 for strategy B which is similar to the earlier findings in [49] and [50] (see Table 11). However, the cost/DALY averted for strategy B is still higher than that of strategy A, thus, the former is eliminated from the list. The result in Table 12 shows that strategy C averts more DALY at a lesser cost than strategy A. It is

**Table 11. ICER of control strategies in the order of DALY averted.**

| Strategy | DALY averted | Cost of strategy($) | ICER($) |
|---|---|---|---|
| Strategy A | 137,112.37 | 6,461,500.00 | 47.13 |
| Strategy B | 101,039.90 | 3,003,000.00 | 95.88 |
| Strategy C | 169,146.61 | 7,310,000.00 | 63.24 |

**Table 12. Comparison between ICER of strategies A and C in terms of DALY.**

| Strategy | DALY averted | Cost of strategy($) | ICER($) |
|---|---|---|---|
| Strategy A | 137,112.37 | 6,461,500.00 | 47.13 |
| Strategy C | 169,146.61 | 7,310,000.00 | 26.49 |

**Table 13. ICER of control strategies in the order of death averted.**

| Strategy | Death averted | Cost of strategy($) | ICER($) |
|---|---|---|---|
| *Strategy B* | 4,316.20 | 3,229,600.00 | 2,100.55 |
| *Strategy C* | 7,227.90 | 7,391,100.00 | 1,429.23 |

**Table 14. ICER of control strategies in the order of DALY averted.**

| Strategy | DALY averted | Cost of strategy($) | ICER($) |
|---|---|---|---|
| *Strategy B* | 101,042.24 | 3,229,600.00 | 89.73 |
| *Strategy C* | 169,205.14 | 7,391,100.00 | 61.05 |

noteworthy that when DALY is used as health benefit in the computation of ICER, the outcome shows that all the strategies are highly cost effective using per capita GDP based criteria and strategy C is the best to be recommended for policy implementation.

It has been shown that 117 deaths occurred within 21 months in the study area because of snake bites (see, S1 Table). However, these number of deaths could have been prevented by using strategy C as an intervention, which has the minimum implementation cost of US $72,548 in comparison to the sum of US$129,074 and US$262,607 that would be required for the execution of strategies A and B, respectively. Each of these strategies would greatly reduce the number of snakebite induced-deaths in the region. Using SBE averted cases as health benefit, the implementation of strategy A only needs about US$6,461,500 to avert 751,800 cases over a 12-month period (i.e., *US*$8.59 per averted case). The sum of *US*$34,429 is required to implement strategy A in order to avert 4008 SBE cases recorded over 21-month period (see, S1 Table). Therefore, this would serve as a guide to both the government and non-governmental organizations in the northeast Nigeria towards reducing the burden of SBE and its related deaths by 50% before the year 2030.

## Effect of early adverse reaction (EAR) on the cost of control strategy

Suppose that the rates at which individuals on antivenom therapy suffer from EAR are set to zero (i.e. $\alpha_1 = \alpha_2 = 0$). This means that nobody who received an antivenom treatment will develop EAR. Further, if strategy B or C is implemented, the results presented in Tables 13 and 14 unveiled a substantial reduction in cost/death and cost/DALY averted in comparison to the ones obtained in Tables 9 and 11, respectively. Therefore, reducing the incidence of antivenom reactions by increasing its safety will curtail the cost of managing SBE burden in the study area.

## Conclusion

A new mathematical model for studying the dynamics of snakebite envenoming (SBE) in a given population is proposed. In the model, treatment and public health enlightenment campaign against SBE are considered as control strategies. Furthermore, the model considered some epidemiological characteristic of snakebite as one of the neglected tropical diseases. The model is fitted using real reported data on snakebite collected from snakebite treatment and research hospital (STRH) Kaltungo, Nigeria. The main findings of the study are as follows:

1. The assessment of public health awareness of susceptible population revealed that the control strategies are significant in controlling the disease in terms of averting the number of SBE cases, deaths and disabilities in the community.

2. If at least 50% of SBE patients received early treatment more than 900 cases of death will be averted in the study area.

3. The implementation of the control strategies either separately or in combination will help in reducing the number of death in the community. However, the combination of the two control strategies averts more than 7,227 deaths and 169,205 DALYs in the population. Given the synergistic reduction in the cost per DALY averted with the two interventions implemented simultaneously compared to when only antivenom is used, the annual amount of US$51–66 million needed to halve the burden in Sub-Saharan Africa (SSA) using antivenom solely [48] will also lessen substantially when antivenom therapy is implemented together with other control measures.

4. The cost effectiveness analysis showed that combination of the two strategies is the most cost effective way of handling SBE in the study area.

5. The sum of US$262,607, US$129,074 and US$72,548 are, respectively, required for each to avert 117 deaths or 2,739 DALYs. Also the sum of US$34,429 is needed in order to avert 4008 SBE cases by using public health enlightenment campaign as an intervention.

6. Early adverse reaction has significant impact on the cost per death and cost per DALY averted when an antivenom is used for treatment separately or in combination with public health enlightenment campaign against SBE.

To the best of the authors' knowledge, this is the first time a study mixing epidemiological modeling with optimal control is applied to snakebite. The model can also be used to assess snakebite envenoming in other settings or countries of high incidence.

## Limitation of the study

This study has the following limitations. Our model could be extended to include some other important epidemiological and demographic features like the spatial and temporal dimensions of human-snakes interaction and also the impact of seasonality on the dynamics of envenoming. Furthermore, the data we collected did not capture unreported cases in the study area and the total population of saw scale viper was estimated not counted. The assumption that only saw scale viper is considered could be relaxed to include multiple species of snakes provided the relevant data could be obtained. Also more control variables could be incorporated into the model to take care of other possible control measures like use of snake repellent, etc.

## Supporting information

**S1 File. Model description.**
(PDF)

**S2 File. Existence of an optimal control.**
(PDF)

**S1 Table. Monthly reported data on SBE collected from treatment and research Hospital Kaltungo, Gombe State, Nigeria (from January, 2019—September, 2020.**
(PDF)

**S1 Fig. Graphs showing the control profile of optimal control strategies.**
(TIF)

## Acknowledgments

We would like to acknowledge, with thanks, the support of Dr. Mukaila Abdullahi for designing the map of the study area. Also, special thanks to Dahiru Ibrahim Sajoh of Modibbo Adama University of Technology, Nigeria for his help in the model simulations.

## Author Contributions

**Conceptualization:** Shuaibu Ahijo Abdullahi, Abdulrazaq Garba Habib, Nafiu Hussaini.

**Data curation:** Shuaibu Ahijo Abdullahi, Abdulrazaq Garba Habib, Nafiu Hussaini.

**Formal analysis:** Shuaibu Ahijo Abdullahi, Nafiu Hussaini.

**Investigation:** Shuaibu Ahijo Abdullahi, Nafiu Hussaini.

**Methodology:** Shuaibu Ahijo Abdullahi, Nafiu Hussaini.

**Project administration:** Nafiu Hussaini.

**Software:** Shuaibu Ahijo Abdullahi, Nafiu Hussaini.

**Supervision:** Abdulrazaq Garba Habib, Nafiu Hussaini.

**Validation:** Shuaibu Ahijo Abdullahi, Nafiu Hussaini.

**Visualization:** Shuaibu Ahijo Abdullahi.

**Writing – original draft:** Shuaibu Ahijo Abdullahi.

**Writing – review & editing:** Abdulrazaq Garba Habib, Nafiu Hussaini.

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
