## [Decision Letter · Decision Letter 0]

23 Mar 2021

Dear Dr. Hussaini,

Thank you very much for submitting your manuscript "Control of snakebite envenoming: a mathematical modeling study" for consideration at PLOS Neglected Tropical Diseases. As with all papers reviewed by the journal, your manuscript was reviewed by members of the editorial board and by several independent reviewers. In light of the reviews (below this email), we would like to invite the resubmission of a significantly-revised version that takes into account the reviewers' comments. 

We cannot make any decision about publication until we have seen the revised manuscript and your response to the reviewers' comments. Your revised manuscript is also likely to be sent to reviewers for further evaluation.

Sincerely,

Adly M.M. Abd-Alla, Prof asso.

Associate Editor

Abdallah Samy

Deputy Editor

Reviewer's Responses to Questions

**Key Review Criteria Required for Acceptance?**

**Methods**

-Are the objectives of the study clearly articulated with a clear testable hypothesis stated?

-Is the study design appropriate to address the stated objectives?

-Is the population clearly described and appropriate for the hypothesis being tested?

-Is the sample size sufficient to ensure adequate power to address the hypothesis being tested?

-Were correct statistical analysis used to support conclusions?

-Are there concerns about ethical or regulatory requirements being met?

Reviewer #1: The objectives of the study are clearly articulated. The study design is generally appropriate, with some concerns indicated below. The population selected for this model analysis is adequate, and the statistical mathematical analysis seems appropriate as well. As this is a mathematical modeling study, there are no ethical concerns.

The authors should consider the following aspects in their methodology:

-The terms ‘Enlighted’ or ‘enlightening’ are not the most appropriate to describe the effect of public education campaigns in snakebite envenoming. The authors may consider using alternative terms such as ‘awareness’ or ‘instructed’. The point here is to underscore the effect of these campaigns in the general awareness or information at the community level. In my view ‘enlightening’ has a different meaning. This is presented to the authors for their consideration.

-Model formulation: When defining the human population, nine categories are proposed and added in the formula. However, the same person may be included in two or more different categories, for example somebody that was ‘enlightened’ and at the same time receive early antivenom treatment. Thus, addition of these categories may provide a mistaken number depending on whether one person may be in two or more categories. 

-Method: Control efforts for educating susceptible individuals: the parameter here is whether an individual or groups of individuals were ‘enlightened’ by providing information. But the doubt remains as how to know whether the persons that received the information indeed learned the concepts transmitted and is now ‘enlightened’. It seems that some sort of evaluation of the knowledge acquired should be introduced in the analysis as to provide a follow up of the effective assimilation of the information provided, i.e., a post-hoc evaluation.

Fig 2: To help the readers locate the region of study in the map of Nigeria, it is suggested that this map includes also the whole map of Nigeria, highlighting the study are.

In the description of the ‘enlightening’ campaigns, only two modalities are considered, i.e., broadcasting material and mobile technology. However, owing to the variety of cultural settings in Nigeria and similar countries it seems that there should be more options for ‘enlightening’ campaigns, such as presential activities in communities, organization of focus groups in schools and other modalities that would adapt to the local contexts. This will increase the cost of these campaigns but will ensure a better communication landscape. The authors may want to include these issues in the discussion of the design and the analysis pf the model proposed.

The model of Bravo et al. considers the density of the snake population. This variable is not used in the currently proposed model for Nigeria. The authors may want to comment on this, especially since there might be regions in their analysis with a different density of snake populations.

Reviewer #2: -Are the objectives of the study clearly articulated with a clear testable hypothesis stated?

The study main objective is to apply epidemiological models to then perform optimal control to reduce burden of snakebite mortality and incidence taking into account the cost of performing prevention strategies and improving antivenom delivery.

-Is the study design appropriate to address the stated objectives?

Authors used an adequate epidemiological model, but the box diagram must be improved. It looks tight and is difficult to read. Then, authors adjusted the model by using parameters from literature and by fitting other parameters with MCMC, which is an adequate methodology. They MUST explain why they assumed the saw scaled viper as the only venomous snakes in their total venomous snakes population. Finally, their optimal control strategy is clear, but they MUST explain deeply the objective function, and why they summed weighted incidence and mortality with their strategies (These variables doesn't have the same dimensions, so the weight variables B1, B2 and Ci must be explained deeplier).

-Is the population clearly described and appropriate for the hypothesis being tested?

They used as a case study northeast Nigeria, a place that has a high burden of snakebite, so the study area and population is appropiate.

-Is the sample size sufficient to ensure adequate power to address the hypothesis being tested?

The sample size is not explicit, because it is not clear if they worked with dissagregated data from the six states of northeast Nigeria, or if they aggergated data. This fact must be explained deeplier.

-Were correct statistical analysis used to support conclusions?

MCMC confidence intervals aseems adequate, but authors do not talk about the convergence of the chains of the algorithm. Maybe using a gelman-rubin diagnostic. Also, the optimal control strategy doesn't have the evidence of the convergence. 

-Are there concerns about ethical or regulatory requirements being met?

No.

**Results**

-Does the analysis presented match the analysis plan?

-Are the results clearly and completely presented?

-Are the figures (Tables, Images) of sufficient quality for clarity?

Reviewer #1: The analysis presentes match well with the analysis plan. The results are clearly and completely presented. The figures and tables could be improved along the lines indicated below.

Effects of public health enlightenment campaigns: In addition to the objective criteria of reduction in SBE as an outcome of the enlightening campaigns, it would be interesting to consider evaluation of improvement in knowledge on how to prevent snakebites through instruments that follow up the knowledge acquired by people that attended or benefited from the campaigns.

The variables ‘enlightening’ and early treatment of snakebites are considered as separate parameters in the analysis. However, there seems to be a clear link between them, since it is likely that people that benefit from the enlightening campaigns would also be people that would procure an early attention to snakebites, assuming that they will not look for traditional treatments that delay the access to health facilities.

Table 14: The meaning of the heads of each column should be indicated in a foot note, i.e., the meaning of I, CI, TE, CTE, etc.

Reviewer #2: -Does the analysis presented match the analysis plan?

Results are great, graphics could be improved in aesthetics, but their contests match the analysis plan. Just as said before, authors must go deeplier into the convergence of MCMC chains and their optimal control strategy.

-Are the results clearly and completely presented?

The paper has a strongly mathematical-biased language, which must be changed to be more comprehensible for the multi-disciplinar audience of PLOS NTDS. Even so, results and conclusions fits totally the requirements of originality, importance and rigurous methodology required by the journal.

-Are the figures (Tables, Images) of sufficient quality for clarity?

Images have a high-quality methodology and importance behind them, but they are not clearly explained and their aesthetics can be improved. The box-diagram of the epidemiological model seems of bad quality, and it looks tight. The figures 3 to 5 must be explained beeter in the image caption: What do you want to point with these plots? Do the model fit well de data? What is the impact of the level of education? What means these strategies A and B in terms of the results shown in Figure 5, what strategy is better? Are necessary the control profiles, or they can go to supplementary materials? These profiles must be explained deeplier in the caption of the figure.

**Conclusions**

-Are the conclusions supported by the data presented?

-Are the limitations of analysis clearly described?

-Do the authors discuss how these data can be helpful to advance our understanding of the topic under study?

-Is public health relevance addressed?

Reviewer #1: The conclusions are generally supported by the data and analyses done, although there are some issues indicated above that need to be considered.

The authors may consider to include, at the end of their manuscript, a section discussing the limitations of the analysis.

The method proposed and described has evident implications from the public health perspective, as it provides interesting tools to model issues related to snakebite envenoming and ways to reduce their impact, including cost-effectiveness analysis.

Reviewer #2: -Are the conclusions supported by the data presented?

Yes, but they could be more direct: ii) What is that substancial number of deaths reduced by that strategy? iii) How many dalys and deaths will be reduced by combining strategies? 

-Are the limitations of analysis clearly described?

No. They don't state the limitations of only using population dynamics of one venomous snake, or the possible effect of seasonality and temporal dynamics of the envenoming, or the limitations of the assumptions behind the proposed model, or what happen with underreporting. A limitations section must be added.

-Do the authors discuss how these data can be helpful to advance our understanding of the topic under study?

-Is public health relevance addressed?

Not so much. This work has a GREAT potential, but that potential is not stated directly. Maybe at the end of conclusions you should add that is the first time that a study mixing epidemiological modeling with optimal control is applied to snakebite, and how can this framework be extrapolated to other countries, and how to use it with fragmented or incomplete datasets.

**Editorial and Data Presentation Modifications?**

Reviewer #1: (No Response)

Reviewer #2: 1. They MUST explain why they assumed the saw scaled viper as the only venomous snakes in their total venomous snakes population. Finally, their optimal control strategy is clear

2. They MUST explain deeply the objective function, and why they summed weighted incidence and mortality with their strategies (These variables doesn't have the same dimensions, so the weight variables B1, B2 and Ci must be explained deeplier).

3. The sample size is not explicit, because it is not clear if they worked with dissagregated data from the six states of northeast Nigeria, or if they aggregated data. This fact must be explained deeplier.

4. Convergence of MCMC chains and optimization must be stated clearly.

5.The paper has a strongly mathematical-biased language, which must be changed to be more comprehensible for the multi-disciplinar audience of PLOS NTDS.

6. Improve figures: The box-diagram of the epidemiological model seems of bad quality, and it looks tight. The figures 3 to 5 must be explained beeter in the image caption: What do you want to point with these plots? Do the model fit well de data? What is the impact of the level of education? What means these strategies A and B in terms of the results shown in Figure 5, what strategy is better? Are necessary the control profiles, or they can go to supplementary materials? These profiles must be explained deeplier in the caption of the figure.

7. Authors must add a limitation section on discussion: Why only use population dynamics of one venomous snake species, what about seasonality and temporal dynamics of the envenoming, which are the limitations of the assumptions behind the proposed model, what happen with underreporting.

8.This work has a GREAT potential, but that potential is not stated directly. Maybe at the end of conclusions you should add that is the first time that a study mixing epidemiological modeling with optimal control is applied to snakebite, and how can this framework be extrapolated to other countries, and how to use it with fragmented or incomplete datasets.

**Summary and General Comments**

Reviewer #1: Mathematical modeling is not my main area of expertise and, therefore, there might be aspects of the development of the model that could be improved and escape my analysis. However, this study is valuable in the sense that a mathematical model approach to assess snakebite envenoming in countries of high incidence, like Nigeria, are badly needed and, in that sense, this contribution is welcomed. The model described could be extrapolated to other settigns and countries. I have a number of concerns with this study which were explained above.

Reviewer #2: The study uses strong mathematical tools to fit epidemiological models to public health data, and then they use these models to perform optimal control based on 2 strategies of prevention and treatment availability. This is a novel approach in snakebite, where these strategies has been proposed but its effect has not been quantified. The significance in snakebite is high, because based on these studies finally the proposed strategies can be evaluated and included into public health scope. 

The paper have weaknesses: The language is directed to a mathematic public, which is not the only public of plos NTDS. This must be changed. Also, there are some data that is missing: Which is the spatial scale that they used? An aggregated for northeast Nigeria? Or the states of this areas? Did the MCMC chains converged? How they assure that they find the optimum? Also, the figures can be improved to fullfil the aesthetics required in a high-impact journal as PLOS NTDS. Even so these weaknesses exist, if the authors fix them the study will be suitable to be published in the journal, because authors did a great job.

PLOS authors have the option to publish the peer review history of their article (what does this mean?). If published, this will include your full peer review and any attached files.

Reviewer #1: No

Reviewer #2: Yes: Carlos Andres Bravo-Vega
---

## [Decision Letter · Decision Letter 1]

26 Jun 2021

Dear Dr. Hussaini,

Thank you very much for submitting your manuscript "Control of snakebite envenoming: a mathematical modeling study" for consideration at PLOS Neglected Tropical Diseases. As with all papers reviewed by the journal, your manuscript was reviewed by members of the editorial board and by several independent reviewers. The reviewers appreciated the attention to an important topic. Based on the reviews, we are likely to accept this manuscript for publication, providing that you modify the manuscript according to the review recommendations. 

Sincerely,

Adly M.M. Abd-Alla, Prof asso.

Associate Editor

Abdallah Samy

Deputy Editor

Reviewer's Responses to Questions

**Key Review Criteria Required for Acceptance?**

**Methods**

-Are the objectives of the study clearly articulated with a clear testable hypothesis stated?

-Is the study design appropriate to address the stated objectives?

-Is the population clearly described and appropriate for the hypothesis being tested?

-Is the sample size sufficient to ensure adequate power to address the hypothesis being tested?

-Were correct statistical analysis used to support conclusions?

-Are there concerns about ethical or regulatory requirements being met?

Reviewer #1: In this revised version the authors have adequately addressed the comments to the methodology.

Reviewer #2: The article improved a lot! Great job. There are still minor issues that should be addressed easily. 

The table of the parameters in the methodology does not have units. Please add them similar to the table of the estimated parameters in results.

Why does u1 and u2 are squared in the objective function if both variables are between 0 and 1? It is not totally clear why do you want to have a cost function non linear.

Why does the factor of u1 and u1 in the objective function have no weight? (Or weight =1?) Please explain this.

The block diagramm quality and map of study area is low, it pixelates when zoom is applied. Please upload a higher resolution version.

**Results**

-Does the analysis presented match the analysis plan?

-Are the results clearly and completely presented?

-Are the figures (Tables, Images) of sufficient quality for clarity?

Reviewer #1: (No Response)

Reviewer #2: Why is the cost function of Cost-effectiveness analysis different than the one presented in the methodology (Objective function?) This function does not have the squares that are present in the method cost function. Please explain deeplier this new function and its differences with cost function of the methodology.

Please explain deeplier how did you computed the DALYS.

**Conclusions**

-Are the conclusions supported by the data presented?

-Are the limitations of analysis clearly described?

-Do the authors discuss how these data can be helpful to advance our understanding of the topic under study?

-Is public health relevance addressed?

Reviewer #1: The conclusions are convincing and supported by the results.

Reviewer #2: Please state more limitations of the model, as which limitations must be taken into account for its appliaction in other countries? In terms of the assumptions, the species, the available data needed to calibrate the model and perform optimal control, and how can be these limitations overcomed.

**Editorial and Data Presentation Modifications?**

Reviewer #1: I do not have further suggestions for modifications

Reviewer #2: Please increase the resolution of the Figure 1 and 2.

Plase upload the code for the study to any data repository recommended by the journal.

**Summary and General Comments**

Reviewer #1: (No Response)

Reviewer #2: The study, as I've said before, has a great potential because it is the first time that optimal control and epidemiological modeling has been applied to snakebite.

PLOS authors have the option to publish the peer review history of their article (what does this mean?). If published, this will include your full peer review and any attached files.

Reviewer #1: No

Reviewer #2: Yes: Carlos Bravo-Vega

Figure Files:

Data Requirements:

Reproducibility:

References

---

## [Editor Report · Decision Letter 2]

5 Aug 2021

Dear Dr. Hussaini,

We are pleased to inform you that your manuscript 'Control of snakebite envenoming: a mathematical modeling study' has been provisionally accepted for publication in PLOS Neglected Tropical Diseases.

Best regards,

Adly M.M. Abd-Alla, Prof asso.

Associate Editor

Abdallah Samy

Deputy Editor

---

## [Editor Report · Acceptance letter]

20 Aug 2021

Dear Dr. Hussaini,

We are delighted to inform you that your manuscript, "Control of snakebite envenoming: a mathematical modeling study," has been formally accepted for publication in PLOS Neglected Tropical Diseases.

Best regards,

Shaden Kamhawi

co-Editor-in-Chief

Paul Brindley

co-Editor-in-Chief
